# 2-Group Symmetries and M-Theory

Michele Del Zotto [1],  Iñaki García Etxebarria [2],  Sakura Schäfer-Nameki [3]

[1] *Mathematics Institute, Uppsala University,*
*Box 480, SE-75106 Uppsala, Sweden*

[1] *Department of Physics and Astronomy, Uppsala University,*
*Box 516, SE-75120 Uppsala, Sweden*

[2] *Department of Mathematical Sciences,*
*Durham University, Durham, DH1 3LE, United Kingdom*

[3] *Mathematical Institute, University of Oxford,*
*Andrew-Wiles Building, Woodstock Road, Oxford, OX2 6GG, UK*

Quantum Field Theories engineered in M-theory can have 2-group symmetries, mixing 0-form and 1-form symmetry backgrounds in non-trivial ways. In this paper we develop methods for determining the 2-group structure from the boundary geometry of the M-theory background. We illustrate these methods in the case of 5d theories arising from M-theory on ordinary and generalised toric Calabi-Yau cones, including cases in which the resulting theory is non-Lagrangian. Our results confirm and elucidate previous results on 2-groups from geometric engineering.

# 1   Introduction

A modern understanding of symmetries characterizes them as a subsector of topological[1] operators with support of various codimensions [1, 2]. The codimension in spacetime determines the dimensions of the charged (not necessarily topological) extended operators in the theory. If a quantum field theory (QFT) has a symmetry described by topological operators of different codimension, these can have non-trivial fusion rules, which mix defects of different codimension. In the most general case, the resulting fusions give rise to categorical symmetries [3–8]. Sometimes, categorical symmetries organize in so-called higher-groups [1,4,9–12]. These are interesting mathematical structures that are ubiquitous in the landscape of quantum field theories:[2] 2-groups have been found in the most basic textbook examples of 4d gauge theories, such as Quantum Electro-Dynamics (QED) and gauge theories with matter [10, 18, 19], as well as in some of the most exotic systems, such as superconformal field theories (SCFTs) in 5d [20] and 6d [19], and little string theories [12, 21].

Since the generic supersymmetric QFT does not admit a conventional Lagrangian description, it is paramount to develop tools in string theory to detect and study the features of such generalized symmetries in the context of various geometric engineering scenarios [19–45].[3] Indeed, 2-groups

---

[1] Meaning that their dependence on their support is topological (which ensures their conservation) up to collisions with charged operators.

[2] We refer our readers to the important foundational papers [13–17] where the crucial interplay among the Green-Schwarz mechanism and 2-groups (and string 2-Lie algebras) was originally derived.

[3] Following the geometric engineering paradigm of exploiting geometries to inform SQFTs [46, 47].

structures have been computed in the literature from geometries engineering SCFTs and LSTs in various dimensions [20, 21, 32, 43, 48]. Although so far the cases of interest to us have only been studied on a resolved phase of the geometry, it seems natural to expect that 2-groups should be an intrinsic feature of these setups, encoded in the geometry of the singularity itself and not only in specific properties of its resolutions, rulings, or deformations. This fact, together with the idea that the charge lattice of extended operators in stringy constructions can be identified with relative homology, indicates that higher groups must be captured by properties of the link (i.e. boundary) of the singularity. For higher-form symmetries this expectation is confirmed by e.g. analysing the non-commutativity of fluxes at the link [24–26, 29, 33, 49–51], the structure of the symmetry TFTs arising from a reduction on the boundary of the compactification [34], and holographic analysis [32, 52, 53].

In this paper we show that this is also the case for 2-groups by deriving the 2-group symmetry from a boundary perspective. Concretely, for geometries where the zero-form symmetries that act faithfully are manifestly realized in terms of non-compact singularities, the latter give rise to singularities in the link geometry. Our main result is a derivation of the 2-group structure from the geometry of the singular link, by relating it to the structure of line operators as described in [18, 20, 48]. We stress that this result is true in general, irrespective of the dimensionality of the singularity in question. We first develop this dictionary in general, and then apply the resulting formalism to the case of M-theory compactifications on three dimensional Calabi-Yau (CY) singularities, which are dual to $(p, q)$ five-brane webs. We also consider generalized toric models, which are dual to webs with multiple 5-branes ending on a single 7-brane. For many of these models the 5d SCFTs are known to have 2-group symmetries [20], and we confirm (and in part extend) these results, using this boundary perspective. Clearly a more general analysis of 2-groups for all 5d SCFTs beyond the (generalized) toric models can be carried out. It requires a framework where the flavor symmetry is manifest, either in terms of gluing of surfaces [54, 55], non-flat resolutions [56–59] or orbifolds [41, 43, 60, 61]. For instance, the case of the $\mathbb{F}_0$ description of the $SU(2)_0$ Seiberg theory is not amenable to this approach, whereas in contrast the $\mathbb{F}_2$ one, which has the manifest non-abelian flavor symmetry realized geometrically, is.

Field theoretically we can characterize higher-groups by studying the background fields for global symmetries that are generated by topological operators. In the case of 2-groups the 1-form symmetry and 0-form symmetry of a QFT satisfy a non-trivial relation: the variation of the background of the 1-form symmetry does not vanish, but depends on the 0-form symmetry background. An alternative but equivalent description of 2-groups emerges by considering equivalence relations on line operators induced by local operators [18, 20, 48]. In particular the 1-form symmetry group is the (Pontryagin dual group to the) group of lines modulo the relation induced by local line changing operators. In the presence of local operators charged under flavor symmetries, this screening picture

can be refined by taking into account the behaviour under the global symmetry of the line changing operator, and considering only those relations induced by operators in (proper) representations of the flavor group.

Field theoretically[4] this interplay between 0- and 1-form symmetry can be detected by computing the charges of local operators under gauge and flavor symmetries. It is this second characterization of the 2-group structure in terms of lines that we reproduce from geometry in this paper. All the various field theoretical ingredients translate nicely in geometrical properties of the boundary of the singularity, and more specifically into the interplay of the local flavor structure arising from the geometry close to the singular locus and the geometry of the rest of the link.

A question that we will not address in this paper is the following: given a 5d theory with a 2-group, one can gauge the 1-form symmetry and obtain a theory with a 0-form symmetry, a 2-form symmetry, and a mixed 't Hooft anomaly connecting them [4]. We expect, based on previous experience in the context of 1-form symmetries (see for instance [24–26, 29, 33, 51, 52]) that both possibilities will be realised in string theory, with the choice being determined by a choice of boundary conditions for fluxes at infinity. Relatedly, we expect that compactification of M-theory on the link of the cone geometry, along the lines of the analysis in [34], leads to a topological field theory – the Symmetry TFT – in one dimension higher. A choice of gapped interface in this theory encodes the polarization choice, between having a 2-group and having ordinary symmetries with a mixed anomaly (this assumes that we can gauge the 1-form symmetry in the 2-group, or the 2-form symmetry in the mixed anomaly, respectively). In this paper we will from the beginning make a choice between these two possibilities by assuming that the non-compact M2-branes lead to genuine line operators in the field theory, which leads to 1-form symmetries in the field theory and therefore 2-group structures. It would certainly be interesting to understand the general situation, but we leave this for the future.

The structure of this paper is as follows: We begin with a brief recap of 2-group symmetries, summarizing their salient features in section 2.1. We then derive our main result, the 2-group symmetries in M-theory geometric engineering from the boundary of the compactification space, in section 2.2. Section 3 applies this general approach to M-theory compactifications on singular Calabi-Yau three-folds to 5d SCFTs, and we show the equivalence to other approaches using five-brane webs and the original intersection theory computations [20]. We provide a flurry of examples, which we discuss using these complementary approaches in section 4.

---

[4] In the context of 5d SCFTs this is however not purely a field-theoretic analysis, since in practice the charges of operators including non-perturbative states such as instanton particles are computed through geometric methods.

## 2 Generalized Symmetries from The Boundary

### 2.1 A Recap of 2-Group Symmetries

In this section we give a brief review of 2-group symmetries following [18, 20, 48] in order to fix notation and conventions. The 2-groups we consider here are built out of discrete 1-form symmetries and continuous 0-form symmetries – for a more general analysis of 2-groups, including continuous 1-form symmetries, see [20, 21, 32, 43, 48]. Consider a theory $\mathcal{T}$ with a discrete 1-form symmetry $\Gamma^{(1)}$ and a continuous 0-form symmetry $\mathcal{F}^{(0)} = F/C$, where $F$ is a simply-connected Lie group, and $C$ a subgroup of its center. We define the global form of the flavor symmetry (0-form) group $\mathcal{F}^{(0)}$ as the group acting faithfully on the spectrum of local operators, or equivalently as the most general structure group that we can choose for the background fields.

The theory $\mathcal{T}$ has a collection of genuine line operators $\mathsf{L}$. We can define two different equivalence relations on $\mathsf{L}$. With this aim in mind let us take two line operators $L_1$ and $L_2$ in $\mathsf{L}$. The first relation, which we denote "$\sim$", asserts that $L_1 \sim L_2$ iff there exists a line changing (0-dimensional) operator between them. This is the equivalence relation used when determining which line operators survive screening, and

$$\widehat{\Gamma}^{(1)} := \mathsf{L}/\!\sim, \qquad L_1 \sim L_2 \Leftrightarrow \exists \text{ local operator } \mathcal{O} \text{ at junction between } L_1 \text{ and } L_2, \qquad (2.1)$$

is the group of lines charged under the 1-form symmetry. Its Pontryagin dual $\Gamma^{(1)} := \mathrm{Hom}(\widehat{\Gamma}^{(1)}, U(1))$ is the group of 1-form symmetries. We can also impose a finer equivalence relation, denoted $\sim'$, which asserts that $L_1 \sim' L_2$ if there exists a line changing operator, transforming in a representation of $\mathcal{F}^{(0)}$, between $L_1$ and $L_2$. We denote the resulting group by

$$\widehat{\mathcal{E}} := \mathsf{L}/\!\sim'. \qquad (2.2)$$

There is a surjective map $\alpha \colon \widehat{\mathcal{E}} \to \widehat{\Gamma}^{(1)}$, since in our definition of $\widehat{\Gamma}^{(1)}$ we did not impose that the line changing operator is in a representation of $\mathcal{F}^{(0)}$, it could be in a representation of $F$ that does not descend to a representation of $\mathcal{F}^{(0)}$. The kernel of this map, $\ker \alpha$, is a subgroup of $\widehat{C}$, where the hat indicates Pontryagin duality: $\widehat{G} := \mathrm{Hom}(G, U(1))$ for any abelian group $G$.[5] In all the examples in this paper $\ker \alpha = \widehat{C}$. Physically, this group can be understood as the group of line operators ending on point operators charged under $C$.

Proceeding in this way, one obtains a short exact sequence of abelian groups

$$0 \to \widehat{C} \longrightarrow \widehat{\mathcal{E}} \xrightarrow{\alpha} \widehat{\Gamma}^{(1)} \to 0, \qquad (2.3)$$

where we have set $\ker \alpha = \widehat{C}$. Our main goal in this paper will be to reformulate this exact sequence in terms of the geometry of the link.

---

[5] In our case both $\Gamma^{(1)}$ and $C$ will be finite abelian groups, so Pontryagin duality gives back the same group, but it is useful to keep the hats on to distinguish between the charged objects (wearing hats) and the associated symmetries (without hats).

It is convenient to dualise (2.3), in which case we have the short exact sequence

$$0 \to \Gamma^{(1)} \to \mathcal{E} \to C \to 0\,. \tag{2.4}$$

The non-trivial extensions are characterized by $\mathrm{Ext}(C, \Gamma^{(1)})$. In the following we will refer to the group $\mathcal{E}$ as the *group extension of $C$ by $\Gamma^{(1)}$*. A given element of $\mathrm{Ext}(C, \Gamma^{(1)})$ determines a Bockstein map

$$\mathrm{Bock}\colon \quad H^n(-; C) \to H^{n+1}(-; \Gamma^{(1)}) \tag{2.5}$$

for the associated long exact sequence in cohomology. Note that Bock is a cohomology operation [62], and is therefore an element of $H^{n+1}(K(C, n); \Gamma^{(1)})$, where $K(C, n)$ is the $n$'th Eilenberg-MacLane space for the group $C$. We can show that this group is indeed isomorphic to $\mathrm{Ext}(C, \Gamma^{(1)})$ as follows. The case of interest to us is $n = 2$, but we include a proof valid for $n > 1$. (The $n = 1$ case is standard.) By definition $\pi_i(K(C, n)) = C$ for $i = n$ and zero otherwise. By the Hurewicz theorem $h\colon \pi_{n+1}(K(C, n)) \to H_{n+1}(K(C, n))$ is surjective for $n > 1$. Since $\pi_{n+1}(K(C, n)) = 0$ we have $H_{n+1}(K(C, n)) = 0$, and the existence of an isomorphism $i\colon \mathrm{Ext}(C, \Gamma^{(1)}) \to H^{n+1}(K(C, n), \Gamma^{(1)})$ then follows from the universal coefficient theorem.

We note that in the cases of interest to us in this paper we have $C = \mathbb{Z}_2$ and $\Gamma^{(1)} = \mathbb{Z}_n$, and $\mathrm{Ext}(C, \Gamma^{(1)}) = \mathrm{Ext}(\mathbb{Z}_2, \mathbb{Z}_n) = \mathbb{Z}_n/2\mathbb{Z}_n = \mathbb{Z}_{\gcd(2,n)}$ [62]. So if $n$ is odd there is no non-trivial extension, and therefore no non-trivial 2-group, while if $n = 2k$ we have $\mathrm{Ext}(\mathbb{Z}_2, \mathbb{Z}_{2k}) = \mathbb{Z}_2$. The non-trivial Bockstein operation in this case is $\mathrm{Sq}^1\colon H^n(-; \mathbb{Z}_2) \to H^{n+1}(-; \mathbb{Z}_2)$ composed with the operation $H^n(-; \mathbb{Z}_2) \to H^n(-; \mathbb{Z}_{2p})$ induced by the non-trivial $\mathbb{Z}_2 \to \mathbb{Z}_{2p}$ homomorphism (which is simply multiplication by $p$).

**2-Groups.** The finite group $C$ also participates on a second short exact sequence

$$0 \to C \to F \to \mathcal{F}^{(0)} \to 0\,. \tag{2.6}$$

This is a central extension of $\mathcal{F}^{(0)}$ by $C$, with associated characteristic class $w_2 \in H^2(\mathcal{F}^{(0)}; C)$. The non-triviality of the 2-group is then measured by the class

$$\mathrm{Bock}(w_2) \in H^3(\mathcal{F}^{(0)}; \Gamma^{(1)})\,. \tag{2.7}$$

In the cases of interest to us, both $w_2$ and $H^3(\mathcal{F}^{(0)}; \Gamma^{(1)})$ are non-trivial, and we can compute the value of $\mathrm{Bock}(w_2)$ using the explicit characterisation in terms of $\mathrm{Sq}^1$ given above. As an example, consider the case $\mathcal{F}^{(0)} = SO(3)$ and $C = \Gamma^{(1)} = \mathbb{Z}_2$. If the short exact sequence (2.4) is non-trivial Bock is non-trivial as a cohomology operation. So $\mathrm{Bock} = \mathrm{Sq}^1$, since this is the only non-trivial cohomology operation. On $BSO(3)$ we have, from the Wu formula, $\mathrm{Bock}(w_2) = \mathrm{Sq}^1(w_2) = w_3$, so we have a non-trivial 2-group.

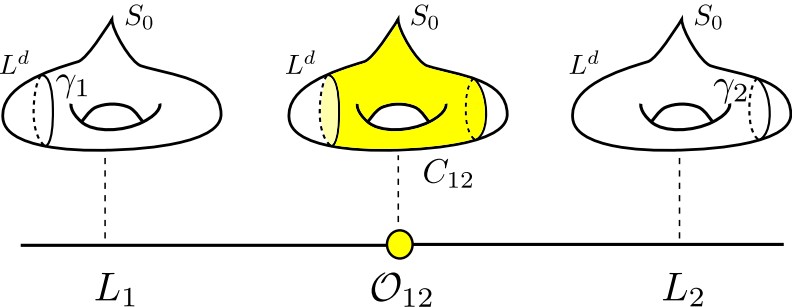

Figure 1: The geometric description of a line changing operator $\mathcal{O}_{12}$ connecting line operators $L_1$ and $L_2$ as a chain (shown in yellow) connecting homologous cycles on the boundary.

In this paper we will first of all determine whether (2.4) splits or not by computing $\widehat{\mathcal{E}}$. From this we can infer also the global form of the flavor symmetry group by taking the quotient

$$\mathcal{F}^{(0)} = \frac{F}{C} \, . \tag{2.8}$$

2-groups of this type were determined for 5d SCFTs with single gauge factors in [20], in 4d $\mathcal{N} = 1$ gauge theories in [18] and in 6d (and more generally for gauge theories with matter in any dimension $d = 3, \cdots, 6$) in [19].

## 2.2  2-Group Symmetries from Link Topology

We now give a geometric realization of (2.3). We will focus on field theories arising from M-theory on singular cones $\mathcal{X}^{d+1}$ with link $\mathbf{L}^d$, geometrically engineering a $(10 - d)$-dimensional field theory $\mathcal{T}_{\mathcal{X}}$.

We focus on the case in which the singularity of $\mathcal{X}^{d+1}$ is not isolated and the non-compact loci supporting the corresponding non-isolated singularities are of dimension $d - 3$. From the point of view of the field theory $\mathcal{T}_{\mathcal{X}}$ the gauge bosons living on the non-compact singular locus in this setting lead to a flavor symmetry. In this paper we assume that the flavor symmetry of the theory $\mathcal{T}_{\mathcal{X}}$ is faithfully reproduced by the geometry of these loci. The non-compact locus of the singularity will intersect $\mathbf{L}^d$ along a subvariety $\mathcal{S}_0$. We denote by $\mathcal{S}$ a small tubular neighbourhood of $\mathcal{S}_0$ inside $\mathbf{L}^d$.

We want to understand the short exact sequence (2.3) from the geometric viewpoint. The geometric interpretation of the group $\widehat{\Gamma}^{(1)}$ is by now standard, and has been studied e.g. in [25,26]. The lines in $\widehat{\Gamma}^{(1)}$ charged non-trivially under 1-form symmetries arise from M2 branes wrapping non-compact 2-cycles which intersect $\mathbf{L}^d$ along representative cycles of non-trivial elements of $H_1(\mathbf{L}^d)$. This group is purely torsional in the cases of interest to us.

It will be illuminating to give an interpretation of this familiar result in terms of line-changing operators, and the equivalence relation $\sim$ defined above in (2.1). Consider two M2 branes wrapping non-compact cycles $\Sigma_1$ and $\Sigma_2$, giving rise to line operators $L_1$ and $L_2$. These cycles will intersect

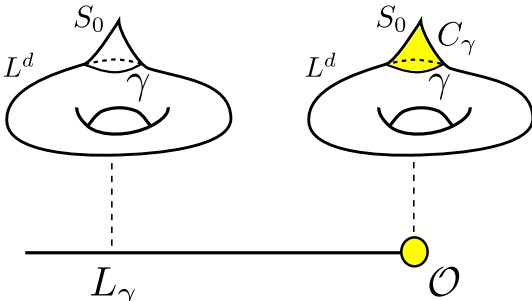

Figure 2: A line operator associated to a homologically trivial curve $\gamma$ on $\mathbf{L}^d$ ending on a point operator. From the point of view of the internal geometry $\mathbf{L}^d$ the point operator at the end corresponds to the M2-brane worldvolume wrapping a chain $C_\gamma$ with boundary $\gamma$.

$\mathbf{L}^d$ on curves $\gamma_1$ and $\gamma_2$. Assume that these two curves are in the same homology class, so there is a chain $C_{12}$ such that $\partial C_{12} = \gamma_1 - \gamma_2$. In this case we can construct a line-changing operator $\mathcal{O}_{12}$ between $L_1$ and $L_2$ by wrapping an M2 brane on the chain $C_{12}$ (times the radial direction). This is shown in figure 1.

This interpretation of the relations in $\sim$ immediately leads to an interpretation of $\widehat{\mathcal{E}}$, and thereby the two-group symmetry. Recall that in this case we want to quotient the space of lines by $\sim'$ in (2.2), which does not include line-changing operators charged under $C$. We can accomplish this from the point of view of the geometry by excising $\mathcal{S}$ from $\mathbf{L}^d$, we denote the resulting space $\mathbf{L}^d - \mathcal{S}$. The chains that pass through the singularity get an extra boundary after excising $\mathcal{S}$, *and therefore no longer lead to relations $\gamma_1 = \gamma_2$ in homology.* We will argue below that this extra boundary encodes the charge under $C$. The surviving homological relations therefore come from chains in $\mathbf{L}^d - \mathcal{S}$, and are precisely those uncharged under the centre of the flavor group. So we identify

$$\widehat{\mathcal{E}} = H_1(\mathbf{L}^d - \mathcal{S}). \tag{2.9}$$

Finally, we can give an interpretation of $\widehat{C}$ in (2.3) along similar lines. Consider a non-compact M2 brane intersecting $\mathbf{L}^d$ along a closed curve $\gamma$ belonging to a non-trivial homology class on $\mathbf{L}^d - \mathcal{S}$ that becomes trivial when embedded in $\mathbf{L}^d$. The associated line operator $L_\gamma$ is therefore trivial as an element of $\widehat{\Gamma}^{(1)}$, because the line can end on a point operator $\mathcal{O}$, given by an M2-brane wrapping a chain $C_\gamma$ with boundary $\gamma$ — see figure 2. These chains will be non-trivial under $\sim'$ if $C_\gamma$ is charged under $C$, which we can detect by computing the class of $\gamma$ in $\operatorname{Tor} H_1(\partial \mathcal{S})$. This mirrors the standard computation of the charge of a line operator under the 1-form symmetries of the field theory summarised above, with the difference that we are viewing the M2 brane wrapped on $C_\gamma$ as a Wilson line for the seven dimensional gauge theory living on the non-compact singular locus (which from the point of view of the 5d SCFT is a flavor sector).

Collecting the results of our discussion so far, we have translated (2.3) into the geometric state-

ment that the following sequence is exact:

$$0 \to \text{Tor}\, H_1(\partial \mathcal{S}) \to H_1(\mathbf{L}^d - \mathcal{S}) \to H_1(\mathbf{L}^d) \to 0\,. \tag{2.10}$$

Although we have derived this exact sequence by reinterpreting the field theory discussion geometrically, it is also possible to derive it via purely geometric arguments as follows. Assume that we have a space $X$, and two subspaces $A, B \subset X$ such that the union of their interiors covers $X$. Then there is a long exact sequence known as the Mayer-Vietoris spectral sequence [62] that reads

$$\ldots \to \tilde{H}_2(X) \to \tilde{H}_1(A \cap B) \to \tilde{H}_1(A) \oplus \tilde{H}_1(B) \to \tilde{H}_1(X) \to \tilde{H}_0(A \cap B) \to \ldots \tag{2.11}$$

where the tildes denote reduced homology groups. For simplicity, in our analysis we will assume that $A$, $B$, $X$ and $A \cap B$ are all connected. (The generalisation is straightforward, but a little cumbersome.) Additionally, we will assume that the boundary map $\tilde{H}_2(X) \to \tilde{H}_1(A \cap B)$ vanishes. We do not have a general argument for this, but it is possible to verify in our explicit examples below that it is the case. With these assumptions in place, the Mayer-Vietoris long exact sequence implies the short exact sequence

$$0 \to H_1(A \cap B) \to H_1(A) \oplus H_1(B) \to H_1(X) \to 0\,. \tag{2.12}$$

We can obtain (2.10) from here by taking $X = \mathbf{L}^d$, $A = \mathbf{L}^d - \mathcal{S}$ and $B = \mathcal{S}'$, where $\mathcal{S}'$ is a slight thickening of the tubular neighbourhood $\mathcal{S}$, so that the interiors of $\mathbf{L}^d - \mathcal{S}$ and $\mathcal{S}'$ indeed cover $\mathbf{L}^d$. The last term clearly is as in (2.10) under this substitution. To recover the other two terms we start by noting that in the geometries analysed below $\mathcal{S} = T \times S^1$, so $\partial \mathcal{S} = \partial T \times S^1$, for some singular toric cone $T$ with boundary $\partial T$ (For instance, in some of the examples below we will have $T$ a neighbourhood of the singular point in $\mathbb{C}^2/\mathbb{Z}_n$, and therefore $\partial T = S^3/\mathbb{Z}_n$, although we emphasise that our analysis is more general). The space $(\mathbf{L}^d - \mathcal{S}) \cap \mathcal{S}'$ deformation retracts to $\partial S = \partial T \times S^1$, so the first term becomes $H_1(\partial T) \oplus \mathbb{Z}$. Singular toric cones of complex dimension 1 and 2 have $H_1(\partial T)$ purely torsional, so we can equivalently write $H_1(\partial S) = \mathbb{Z} \oplus \text{Tor}\, H_1(\partial T)$. For the middle term, we have $H_1(B) = H_1(\mathcal{S}') = H_1(\mathcal{S}) = H_1(T) \oplus H_1(S^1) = \mathbb{Z}$, using that toric varieties have no non-trivial 1-cycles. An explicit analysis of the inclusion map $H_1(A \cap B) \to H_1(A) \oplus H_1(B)$ then shows that it restricts to an isomorphism on the $\mathbb{Z}$ factors comings from the $S^1$ factor in $\mathcal{S}$, so we recover (2.10) also as a mathematical consequence of Mayer-Vietoris.

One technical point to highlight at this stage is that in the derivation above we have used rather special properties of the geometries studied in this paper to conclude that the Mayer-Vietoris long exact sequence splits, so that we end up with a short exact sequence for the groups of interest. In particular, our examples below are such that $\partial T$ is a lens space $S^3/\mathbb{Z}_n$, associated with a flavor algebra $\mathfrak{su}(n)$, and our analysis gives $C = \mathbb{Z}_n$, which agrees with what one finds from field theory

considerations in these cases. In more general situations we do not expect the Mayer-Vietoris sequence to split, but we can still write a tautological short exact sequence

$$0 \to \ker(a) \to H_1(\mathbf{L}^d - \mathcal{S}) \oplus H_1(\mathcal{S}') \xrightarrow{a} H_1(\mathbf{L}^d) \to 0 \,, \tag{2.13}$$

still under the assumption that the $\partial \mathcal{S}$ is connected. In this case we would identify $C = \mathrm{Tor} \, \ker(a)$.

Our main task therefore becomes to compute

$$\widehat{\mathcal{E}} = H_1(\mathbf{L}^d - \mathcal{S}). \tag{2.14}$$

Below we will introduce methods that allow us to compute this group systematically in ordinary and generalised toric varieties, but before doing so let us comment briefly on how this discussion connects to previous work [19, 20]. Consider the long exact sequence for the pair $(\mathcal{X}_\epsilon^{d+1}, \mathbf{L}^d - \mathcal{S})$, where $\mathcal{X}_\epsilon^{d+1}$ is a neighborhood of the origin of the conical singularity $\mathcal{X}^{d+1}$ (so that $\partial \mathcal{X}_\epsilon^{d+1} = \mathbf{L}^d$):

$$\ldots \to H_2(\mathcal{X}_\epsilon^{d+1}) \to H_2(\mathcal{X}_\epsilon^{d+1}, \mathbf{L}^d - \mathcal{S}) \to H_1(\mathbf{L}^d - \mathcal{S}) \to H_1(\mathcal{X}_\epsilon^{d+1}) \to \ldots \tag{2.15}$$

Since $\mathcal{X}_\epsilon^{d+1}$ is a special holonomy variety we expect $H_1(\mathcal{X}_\epsilon^{d+1}) = 0$ (this holds for toric varieties and singular hypersurfaces, for instance; we assume that this is the case for the validity of this analysis), and the sequence terminates:

$$\ldots \to H_2(\mathcal{X}_\epsilon^{d+1}) \xrightarrow{i} H_2(\mathcal{X}_\epsilon^{d+1}, \mathbf{L}^d - \mathcal{S}) \to H_1(\mathbf{L}^d - \mathcal{S}) \to 0 \tag{2.16}$$

which implies

$$\widehat{\mathcal{E}} = H_1(\mathbf{L}^d - \mathcal{S}) = \mathrm{coker}(i) \,. \tag{2.17}$$

This statement can be interpreted in terms of screening, generalising the discussion in [23, 24] to 2-groups: the lines in $H_2(\mathcal{X}_\epsilon^{d+1}, \mathbf{L}^d - \mathcal{S})$ are lines where we keep track of the flavor charge — the fact that we are excising $\mathcal{S}$ from $\mathbf{L}^d$ in the pair means that relative cycles with different flavor charge, that would be equivalent in $H_2(\mathcal{X}_\epsilon^{d+1}, \mathbf{L}^d)$, are no longer equivalent in $H_2(\mathcal{X}_\epsilon^{d+1}, \mathbf{L}^d - \mathcal{S})$, since the chain connecting them does necessarily pass through $\mathcal{S}$ (as in our boundary analysis above).

# 3 2-Group Symmetries in 5d from M-theory

In this section we explain how to exploit the general method discussed in the previous section to recover the results on 2-groups from the geometry of the boundary in the context of 5d SCFTs arising from compactification of M-theory on local CY singularities $\mathcal{X}$. For simplicity, we will mostly focus on cases where $\mathcal{X}$ is a toric CY cone, since in these cases we can apply results from [63] to capture the geometry of the boundary. For toric CY singularities the corresponding 5d SCFTs also

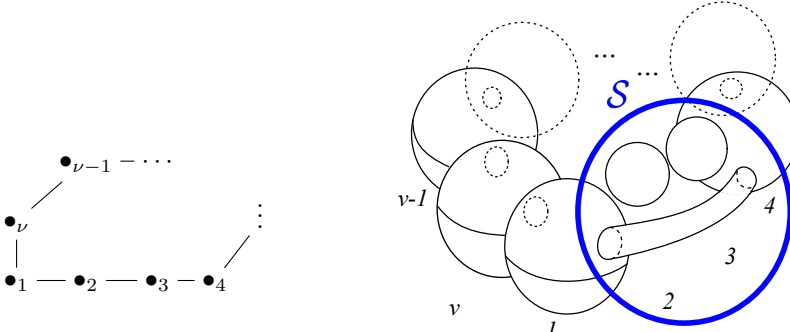

Figure 3: Schematic topology of $\mathcal{C}^3_{\mathbf{L}_{\mathcal{X}}}$ [63] – see also [26] for the case $\mathcal{X}$ non-isolated. In this figure we have 4 vertices along an external edge corresponding to a $\mathbb{C}^2/\mathbb{Z}_3$ singularity. We denote as $\mathcal{S}$ the neighborhood of the singular locus we need excise when considering the 2-group structure – clearly after the excision the Lens spaces $\mathbf{L}_1$ and $\mathbf{L}_4$ in the above figure become contratible and do not contibute to $H_1(\mathbf{L}^5_{\mathcal{X}} - \mathcal{S})$.

have a dual geometric engineering in terms of webs of $(p, q)$ five-branes — this gives rise to a dual IIB version of the excision method which is extremely powerful in practice (see section 3.2).

In particular, in section 3.3 we show the equivalence of the methods presented in this paper with the prescription for determining the 2-group structure uncovered in [20] for the case of CY threefolds $\mathcal{X}$ whose non-compact singularities faithfully reproduce the flavor symmetries of the corresponding 5d SCFTs.

Our analysis in this section proves and extends a proposal in [43]. The authors of that paper consider orbifolds of the form $S^5/\Gamma$ with a normal subgroup $H \triangleleft \Gamma$ acting with fixed points on $S^5$, and proposed that in these cases the 2-group structure is associated with the short exact sequence

$$0 \to H_1(S^5/\Gamma) \to \Gamma^{\mathrm{ab}} \to \Gamma^{\mathrm{ab}}/H_1(S^5/\Gamma) \to 0 \tag{3.1}$$

where $\Gamma^{\mathrm{ab}} \coloneqq \Gamma/[\Gamma, \Gamma]$ denotes the abelianisation of $\Gamma$. We can reinterpret this in terms of our discussion above: this sequence is the Pontryagin dual of (2.10), noticing that $\pi_1(S^5/\Gamma - S) = \Gamma$ (since $\Gamma$ acts freely on $S^5/\Gamma - S$), and by the Hurewicz isomorphism $H_1(S^5/\Gamma - S) = \pi_1(S^5/\Gamma - S)^{\mathrm{ab}}$.

In the section 4.4 below we discuss a possible generalization of our arguments for SCFTs that arise outside of the toric realm.

## 3.1 2-groups from the Boundary: the case of Toric 5d SCFT

Consider the case $\mathcal{X}$ is a toric CY singularity. We will first apply our general singularity excision approach to two-groups in this context and show subsequently the equivalence with the direct intersection computation.

To each such singularity corresponds a toric diagram, a convex polytope embedded in $\mathbb{Z}^2$ with $\nu$ external vertices $\mathbf{v}_i \in \mathbb{Z}^2$, $i = 1, ..., \nu$. In this paper we are interested in the geometry of the link

of $\mathcal{X}$, which we denote $\mathbf{L}_\mathcal{X}^5$. As first argued (to the best of our knowledge) in [63], one has that

$$H_n(\mathbf{L}_\mathcal{X}^5) = H_n(\mathcal{B}_{\mathbf{L}_\mathcal{X}}^3) \qquad \text{for } n \leq 2, \tag{3.2}$$

where $\mathcal{B}_{\mathbf{L}_\mathcal{X}}^3$ is a 3-chain of lens spaces

$$\mathcal{B}_{\mathbf{L}_\mathcal{X}}^3 \simeq \mathbf{L}_{n_1} \veebar \mathbf{L}_{n_2} \veebar \ldots \veebar \mathbf{L}_{n_\nu} \tag{3.3}$$

where $\veebar$ denotes that the lens spaces are joined along their torsion cycle. The $n_i$ are determined as follows. For each external vertex $\mathbf{v}_i$, $i \in \{1, \ldots, \nu\}$, construct the triangle $T_i$ defined by the vertex and the two vertices adjacent to it, that is, the convex hull of $\{\mathbf{v}_{i-1}, \mathbf{v}_i, \mathbf{v}_{i+1}\}$ (with $\mathbf{v}_0 \coloneqq \mathbf{v}_\nu$ and $\mathbf{v}_{\nu+1} \coloneqq \mathbf{v}_1$). Then

$$n_i = 2\text{Area}(T_i). \tag{3.4}$$

Whenever we have a collection of $m+1$ external lattice points $\mathbf{v}_i$ along an edge, this corresponds to the presence of a non-compact curve of singularities $\mathbb{C}^2/\mathbb{Z}_m$, giving a factor $\mathfrak{su}(m)$ of the global symmetry of the 5d SCFT $\mathcal{T}_\mathcal{X}$. If this is the case, the corresponding triangles $T_i$ will have zero area. Since the lens space $\mathbf{L}_n$ is a circle fibration over $\mathbf{S}^2$ of degree $n$ ($\mathbf{L}_n \simeq S^3/\mathbb{Z}_n$ for $n \geq 1$) we can include the case $n = 0$ as $\mathbf{L}_0 \cong \mathbf{S}^2 \times \mathbf{S}^1$ — whenever we have points along an edge, upon crepant resolution the local geometry of the $\mathbf{T}^2$ fiber considered in [63] around the point is that of $\mathbf{S}^2 \times \mathbf{S}^1$ (see figure 3). Additionally, one can show that [63]

$$H_1(\mathcal{B}_{\mathbf{L}_\mathcal{X}}^3) = \mathbb{Z}_{\gcd(n_1, \ldots, n_\nu)}, \tag{3.5}$$

and in the toric case

$$\Gamma^{(1)} \simeq H_1(\mathbf{L}_\mathcal{X}^5) = \mathbb{Z}_{\gcd(n_1, \ldots, n_\nu)} \tag{3.6}$$

taking into account that $\gcd(0, \ldots) = \gcd(\ldots)$.

Now consider the case we have a single non-compact curve of singularities, corresponding to the fact that the flavor symmetry of the 5d SCFT has Lie algebra $\mathfrak{su}(m)$. As discussed above this corresponds to a sequence of outer vertices $\mathbf{v}_{i_1}, \mathbf{v}_{i_2}, \ldots \mathbf{v}_{i_{m+1}}$ which are all aligned along an external edge. Then we can explicitly apply the method outlined in section 2 to recover the 2-group structure for the 5d SCFT at hand. In presence of such a singularity, the link itself will have a singular locus $\mathcal{S}$ and we are interested in computing

$$\hat{\mathcal{E}} = H_1(\mathbf{L}_\mathcal{X}^5 - \mathcal{S}). \tag{3.7}$$

The latter is easily obtained from the discussion in [63]. Removing the neighborhood of the singular locus $\mathcal{S}$ alters the topology of the 3-chain $\mathcal{B}_{\mathbf{L}_\mathcal{X}}^3$, rendering contractible the Lens spaces $\mathbf{L}_{v_{i_1}}$ and

$\mathbf{L}_{v_{i_{m+1}}}$. We denote the resulting 3-chain $\widehat{\mathcal{B}}^3_{\mathbf{L}_\mathcal{X}}$. We can always relabel the outer vertices so that the first $m+1$ corresponds to the $m+1$ aligned ones. Proceeding in this way, we obtain that

$$\widehat{\mathcal{B}}^3_{\mathbf{L}_\mathcal{X}} \sim \mathbf{L}_{m+2} \veebar \mathbf{L}_{m+3} \veebar \cdots \veebar \mathbf{L}_\nu \tag{3.8}$$

where $\sim$ is homotopy equivalence. Then by the same argument that lead to the conclusion in [63], we obtain that

$$\hat{\mathcal{E}} = H_1(\mathbf{L}^5_\mathcal{X} - \mathcal{S}) = H_1(\widehat{\mathcal{B}}^3_{\mathbf{L}_\mathcal{X}}) = \mathbb{Z}_{\gcd(n_{m+2},\ldots,n_\nu)} \,. \tag{3.9}$$

**Example:** $SU(2)_0$. Let us consider the simplest example: the 2-group of the 5d SCFT, which has a Coulomb branch description as $SU(2)_0$ [20]. The toric diagram is

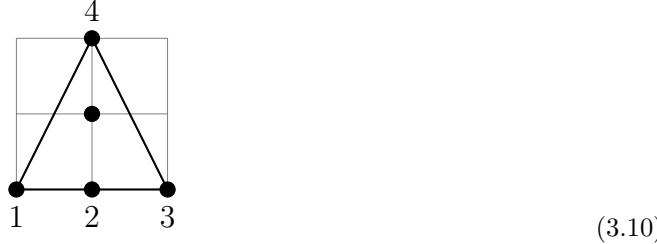

$$\tag{3.10}$$

where we labeled the external vertices. Now we apply the above algorithm to compute the 1-form symmetry by considering the triangles $T_i := \Delta(i-1, i, i+1)$:

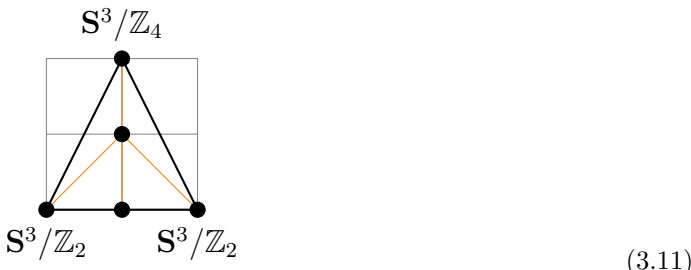

$$\tag{3.11}$$

In orange we show the triangulations. E.g. for the vertex 3 we find $|\Delta(2,3,4)| = n_3 = 2$ etc. Thus the 1-form symmetry is $\Gamma^{(1)} = \mathbb{Z}_{\gcd(2,2,4)} = \mathbb{Z}_2$, as expected for this theory.

To determine $\widehat{\mathcal{E}}$ requires the excision of the flavor nodes. In this example the only edge is the bottom edge, which has the vertex 2 on it. Thus we find

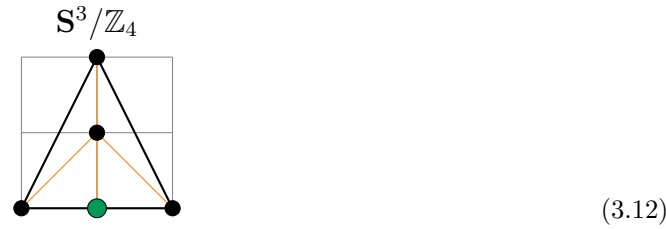

$$\tag{3.12}$$

where the green vertex is the one we excised. The remaining asymptotic topology is $\mathbf{S}^3/\mathbb{Z}_4$ and thus

$$\widehat{\mathcal{E}} = \mathbb{Z}_4 \qquad\qquad (3.13)$$

consistent with the existence of the 2-group [20]. Out of this analysis we also infer the global form of the flavor symmetry for this theory is $SU(2)/\mathbb{Z}_2$, consistent with [64,65].

The above discussion can be generalized to the cases where the singularity $\mathcal{X}$ has several non-compact singularities, each corresponding to a factor $\mathfrak{su}(m_k)$ of the flavor symmetry. It is natural to consider each of these factors separately, excising only the neighborhood $\mathcal{S}_k$ of the corresponding singular locus on $\mathbf{L}_{\mathcal{X}}^5$. The analysis is the same we discussed above, and each of these factors would end up forming a different extension with the one form symmetry. For an explicit example of this see section 4.2 below.

**Summary of the Computational Approach.** To summarize we find the following computational description for the 1-form symmetry and 2-group for toric geometries.

1. Compute the asymptotic lens space topology for each external vertex of the toric diagram, i.e. for each triple of vertices $v_{i-1}, v_i, v_{i+1}$ compute the volume of the triangle $\Delta(v_{i-1}, v_i, v_{i+1}) = n_i$, then the boundary topology is $S^3/\mathbb{Z}_{n_i}$. The 1-form symmetry is then

$$\Gamma^{(1)} = \mathbb{Z}_{\gcd(n_1, \cdots, n_N)}. \qquad\qquad (3.14)$$

   where $N$ is the total number of external vertices. Note that vertices along edges get assigned $S^3/\mathbb{Z}_0 \sim S^2 \times S^1$.

2. To determine $\mathcal{E}$ consider the vertices along edges. For each edge $e_\ell$, let

$$V_\ell = \{v_i : \ v_i \in e_\ell \text{ and } \ v_i \cap (\partial e_\ell) = \emptyset\} \qquad\qquad (3.15)$$

   be the vertices along the edge, however not including the corners. Note that these correspond to non-compact divisors, which generate a flavor symmetry algebra $\mathfrak{f} = \mathfrak{su}(|V_\ell| + 1)$. For each edge we excise $V_\ell$ and $\partial e_\ell$ and compute

$$\mathcal{E}_\ell = \mathbb{Z}_{\gcd(\{n_i : \ v_i \notin e_\ell\})}, \qquad\qquad (3.16)$$

   i.e. we excise the lens spaces along the edge $e_\ell$ including the corners. Then the associated flavor symmetry group is

$$\mathcal{F}_\ell = \frac{SU(|V_\ell| + 1)}{C_\ell}, \qquad C_\ell = \frac{\mathcal{E}_\ell}{\Gamma^{(1)}}. \qquad\qquad (3.17)$$

   If this group is a a non-trivial extension of $\Gamma^{(1)}$

$$1 \to \Gamma^{(1)} \to \mathcal{E}_\ell \to C_\ell \to 1, \qquad\qquad (3.18)$$

then there is a non-trivial 2-group if in addition there is non-zero Postnikov class in

$$\text{Bock}(w_2) = w_3 \in H^3(B\mathcal{F}, \Gamma^{(1)}) \,. \tag{3.19}$$

Repeating this analysis along all edges $e_\ell$ results in the full symmetry structure of the theory, identifying which 0-form symmetry factors participate in the 2-group structure.

## 3.2 Excision and $(p, q)$-Fivebrane Webs

In the dual 5-brane webs the description becomes even simpler – and combinatorially easy to implement. Consider a toric polygon for a Calabi-Yau three-fold, and let $W = \{(p_i, q_i)\}$ be the dual labels for the 5-brane web, i.e. the differences of consecutive external vertices in the polygon. The precise relation is that for a $\mathbf{v}_i$ and $\mathbf{v}_{i+1}$ consecutive external vertices

$$\mathbf{v}_i - \mathbf{v}_{i+1} = (a, b, 0) \qquad \Leftrightarrow \qquad (p, q) = (b, -a) \,. \tag{3.20}$$

In this convention, the D5-branes are horizontal (i.e. charge $(1, 0)$) and NS5 vertical (i.e. $(0, 1)$). Notice that here it is key to include also vertices along a single edge (i.e. $(p, q)$ charges will have multiplicities). This duality between toric diagrams and $(p, q)$ webs has a beautiful gemetrical interpretation [47, 66] – see figure 4. The diagram describing the positions of the $(p, q)$ fivebranes is dual in geometry to a fibration of a $T^2$ over a plane. The $(p, q)$ segments can be understood as characterising loci where a 1-cycle $pA + qB$ of the $T^2$ is shrinking to zero size. In presence of $m$ consecutive parallel (p,q) fivebranes extending at infinity we see that in the dual description we find a resolved ALE singularity of type $\mathbb{C}^2/\mathbb{Z}_m$ whose locus is a non-compact curve. On the contrary, the components of the boundary that are dual to regions between $(p, q)$ fivebranes that are non-parallel correspond to the lens-spaces, as the corresponding shrinking one-cycles change, giving rise to a Hopf-like fibration description of the space $S^3/\mathbb{Z}_n$. This explains why the topology of the boundary is encoded in the structure of the asymptotic (p,q) fivebranes that extend to infinity. It was indeed shown in [30], that the 1-form symmetry can be computed from $W$ simply by taking the Smith normal form (SNF)

$$\text{diag}(n_1, n_2) = \text{SNF}(W) \,. \tag{3.21}$$

To determine whether there is a non-trivial extension, we need to account for the flavor symmetry groups. As explained above, these are captured by the multiplicities in the $(p, q)$-charges of consecutive parallel $(p, q)$ fivebranes. Denote by

$$m_{(p,q)} = \text{multiplicity of } (p, q) \text{ in } W \tag{3.22}$$

one such multiplicity. In particular this means that the flavor symmetry algebra $\mathfrak{f}$ of the 5d SCFT has a subalgebra $\mathfrak{su}(m_{(p,q)})$. Clearly now we see what is the dual procedure of the excision of

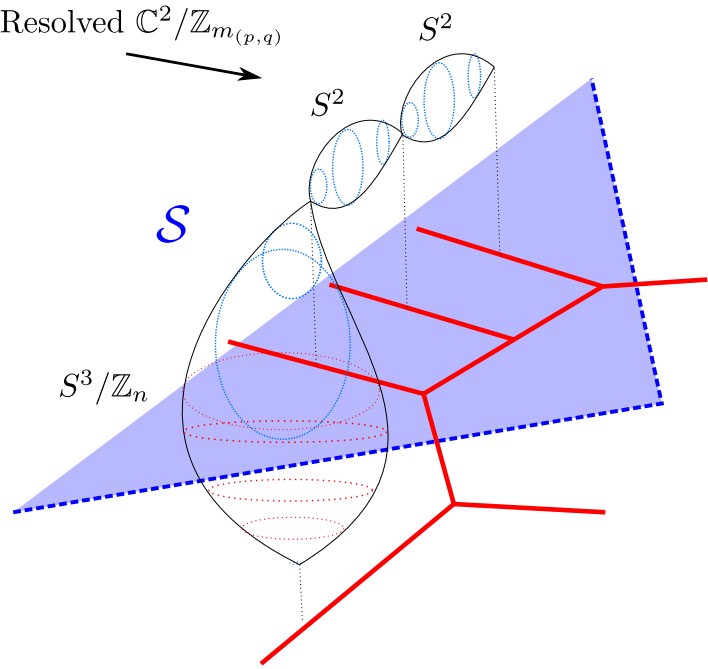

Figure 4: Duality between geometry and $(p, q)$ web. The figure shows the combination of both brane-web and geometry. The location of the $(p, q)$ branes (in red) is dual to loci where $T^2$ cycles degenerate. This gives rise to the topologies we draw above. The excision locus, corresponding to the shaded region inside the wedge, is denoted by $\mathcal{S}$. Removing $\mathcal{S}$ is dual to deleting the corresponding parallel $(p, q)$ branes.

the locus $\mathcal{S}$ corresponding to this singularity of $\mathbf{L}^5_{\mathcal{X}}$: this is dual to removing the corresponding collection of $m_{(p,q)}$ parallel $(p,q)$ webs corresponding to the $\mathbb{C}^2/\mathbb{Z}_{m_{(p,q)}}$ singularity — see Figure 4. Therefore we introduce

$$W^{\text{red}}_{(p,q)} = \text{Matrix obtained by removing } (p,q)^{m_{(p,q)}} \text{ from } W\,. \tag{3.23}$$

The flavor symmetry subalgebra $\mathfrak{su}(m_{(p,q)})$ contributes a non-trivial extension to $\widehat{\mathcal{E}}$ if

$$\text{SNF}(W^{\text{red}}_{(p,q)}) = \text{diag}(N_1, N_2)\,, \tag{3.24}$$

where $N_i/n_i > 1$ and $\gcd(N_i/n_i, n_i) \neq 1$.

**Example:** $SU(2)_0$. Let us apply this again to the $SU(2)_0$ SCFT in 5d, which has the dual brane-web (shown in red):

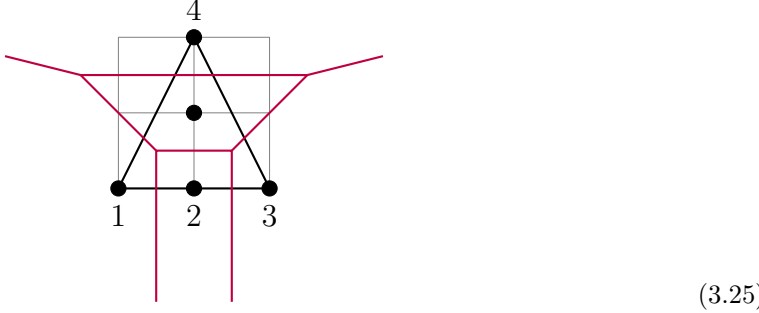

$$\tag{3.25}$$

The set of $(p,q)$ charges is

$$W = \begin{pmatrix} 0 & 1 \\ 0 & 1 \\ -2 & -1 \\ 2 & -1 \end{pmatrix} \tag{3.26}$$

The SNF results precisely in the $\mathbb{Z}_2$ 1-form symmetry. We see that there is only one set of $(p,q)$-charges with multiplicity bigger than 1: $(p,q) = (-1,0)$ and

$$m_{(-1,0)} = 2\,. \tag{3.27}$$

Thus the reduced matrix is

$$W^{\text{red}(-1,0)} = \begin{pmatrix} -2 & -1 \\ 2 & -1 \end{pmatrix} \tag{3.28}$$

whose SNF is $\text{diag}(4,1)$, and thus confirms again the non-trivial extension $\mathcal{E} = \mathbb{Z}_4$, as expected [20].

## 3.3 Equivalence to Intersection-Theoretic Approach to 2-Groups

Here we show the equivalence of our method with the previous results that appeared in the literature about 2-groups [20]. In the context of the CY singularities we are considering in this paper as the main source of examples the key sequence (2.16) reads

$$\ldots \to H_2(\mathcal{X}^6_\epsilon) \xrightarrow{i} H_2(\mathcal{X}^6_\epsilon, \mathbf{L}^5_{\mathcal{X}} - \mathcal{S}) \to H_1(\mathbf{L}^5_{\mathcal{X}} - \mathcal{S}) \to 0 \tag{3.29}$$

where

$$\widehat{\mathcal{E}} = H_1(\mathbf{L}_{\mathcal{X}}^5 - \mathcal{S}) = \operatorname{coker} i \,. \tag{3.30}$$

In this case, we can give a second field theory interpretation of $\widehat{\mathcal{E}}$ which recovers the previous results about 2-groups in the literature. By Lefschetz duality for triples [62], we have $H_2(\mathcal{X}_\epsilon^6, \mathbf{L}_{\mathcal{X}}^5 - \mathcal{S}) = H^4(\mathcal{X}_\epsilon^6, \mathcal{S})$, which by the universal coefficient theorem, and the fact that $H_3(\mathcal{X}_\epsilon^6, \mathcal{S}) = 0$ since $\mathcal{X}$ is toric, is equal to $\operatorname{Hom}(H_4(\mathcal{X}_\epsilon^6, \mathcal{S}), \mathbb{Z})$. The group $H_4(\mathcal{X}_\epsilon^6, \mathcal{S})$ is the group of compact divisors of $\mathcal{X}_\epsilon^6$, together with the singular non-compact divisors, which give rise to relative 4-cycles in the pair $(\mathcal{X}_\epsilon^6, \mathcal{S})$. From this Lefschetz dual viewpoint, the short exact sequence (2.16) becomes

$$\ldots H_2(\mathcal{X}_\epsilon^6) \xrightarrow{q} \operatorname{Hom}(H_4(\mathcal{X}_\epsilon^6, \mathcal{S}), \mathbb{Z}) \to H_1(\mathbf{L}_{\mathcal{X}}^5 - \mathcal{S}) \to 0 \,, \tag{3.31}$$

where the map $q$ is defined by the intersection pairing: $(q(\Sigma_2))(D_4) = \Sigma_2 \cdot D_4$. This intersection pairing therefore measures the gauge and flavor charges of the dynamical states. By exactness

$$\widehat{\mathcal{E}} = H_1(\mathbf{L}_{\mathcal{X}}^5 - \mathcal{S}) = \operatorname{coker} q \,, \tag{3.32}$$

reproducing the prescription introduced for computing 2-groups in [20]. Indeed, the pairing $(q(\Sigma_2))(D_4)$ captures the intersection of compact and non-compact divisors with compact curves, i.e.

$$\mathcal{M} = \left( \begin{array}{c} \mathcal{M}_{4,2}^G \\ \hline \mathcal{M}_{4,2}^F \end{array} \right) \,, \tag{3.33}$$

where the superscript specifies whether these are intersections with compact $(G)$ or non-compact $(F)$ divisors. The fact that we are computing the cokernel of $q$ then reproduces the prescription of [20], namely

$$\widehat{\mathcal{E}} = \mathbb{Z}^{r+f} / \mathcal{M} \mathbb{Z}^{r+f} \,, \tag{3.34}$$

thus showing that the formalism we developed in this paper correctly reproduces the previous results.

# 4 Examples with 2-Group Symmetries

## 4.1 Examples: 5d $SU(N)_k$ theories

For pure gauge theories in 5d with gauge group $SU(N)$ and CS-level $k$ it is known [20] that the theories with 2-groups are

$$SU(2n)_{2n} : \qquad \Gamma^{(1)} = \mathbb{Z}_{2n} \,, \qquad \mathcal{F} = SO(3) \,. \tag{4.1}$$

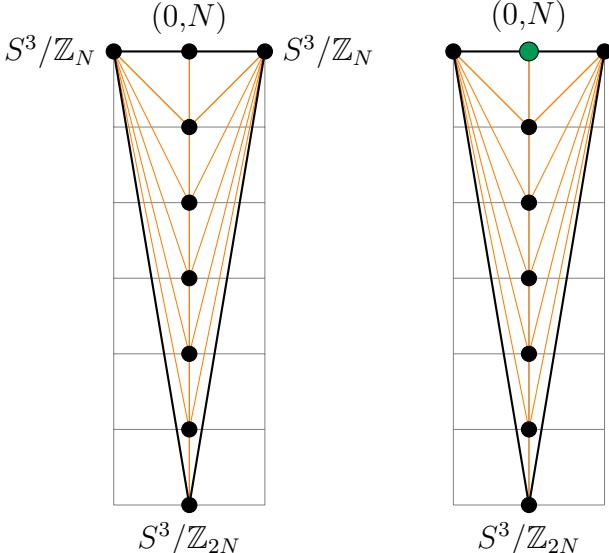

Figure 5: The toric diagram from $SU(N)_N$ shown here on the left for $SU(6)_6$, which has $\mathbb{Z}_6$ 1-form symmetry. The orange lines indicate the triangles which determine the order of the $S^3/\mathbb{Z}_k$ quotient. The right figure shows the toric diagram after we excise the $A_1$ singularity (shown in green), which is associate to the $\mathcal{F} = SO(3)$ flavor symmetry group. The only remaining lens space is the $S^3/\mathbb{Z}_{2N}$. For even $N$ this forms a non-trivial extension with the 1-form symmetry $\mathbb{Z}_N$ and thereby a 2-group.

The theories with other CS-levels or $SU(2n+1)$ have trivial 2-groups. The toric diagram for one of these is e.g. in figure 5. From the left hand figure we infer that the one-form symmetry is indeed $\mathbb{Z}_{\gcd(N,N,2N)} = \mathbb{Z}_N$, whereas from the right hand side, after we excise the flavor symmetry vertex, the only remaining lens space singularity at the boundary is $S^3/\mathbb{Z}_{2N}$. For $N = 2n$ these fit into the non-split short exact sequence

$$1 \; \to \; \mathbb{Z}_{2n} \; \to \; (\mathcal{E} = \mathbb{Z}_{4n}) \; \to \; \mathbb{Z}_2 \to 1 \,. \tag{4.2}$$

We can of course also apply the approach using the dual webs in section 3.2. First of all it is obvious that for $k \nmid N$ there is no 2-group (since there is no non-abelian flavor symmetry, and thus no multiplicities in the $(p,q)$-charge matrix $W$). For $k = N$,

$$W = \begin{pmatrix} 0 & 1 \\ 0 & 1 \\ 1 & -N \\ 1 & N \end{pmatrix} \tag{4.3}$$

The Smith normal form of $W$ is $\mathrm{diag}(1,N)$, and thereby $\Gamma^{(1)} = \mathbb{Z}_N$. To compute the extension group $\mathcal{E}$, note that the entry $(-1,0)$ has multiplicities $m_{(-1,0)} = 2$ and so

$$W^{\mathrm{red}}_{(-1,0)} = \begin{pmatrix} 1 & -N \\ 1 & N \end{pmatrix} \tag{4.4}$$

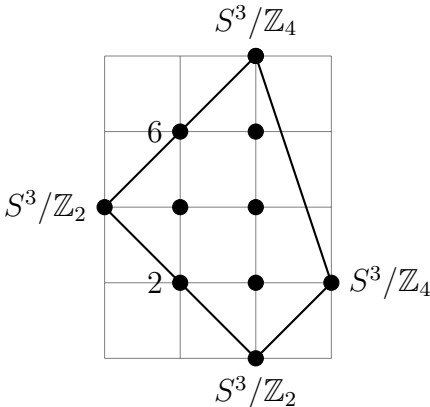

Figure 6: The toric diagram for $SU(2)_0 - SU(4)_0$ quiver.

whose Smith normal form is

$$\text{SNF}(W^{\text{red}}_{(-1,0)}) = \text{diag}\left(1, 2^{(1+(-1)^N)/2}N\right). \tag{4.5}$$

So that $\mathcal{E} = \mathbb{Z}_{2N}$ for $N$ even and $\mathbb{Z}_N$ for $N$ odd.

## 4.2 Quivers

There are numerous quiver theories which have 2-groups. We select out the following class (which are closely related to $SU(N)_k + 1\boldsymbol{AS}$, as we will discuss later). The new feature in this class of examples is that the flavor symmetry has multiple components.

Let us first discuss the simplest example. Consider the toric diagram drawn in figure 6. This theory has a description in terms of the strongly-coupled limit of an $SU(2)_0 - SU(4)_0$ quiver gauge theory. Figure 6 shows already the asymptotic lens spaces, which imply the 1-form symmetry is $\Gamma^{(1)} = \mathbb{Z}_2$.

To compute the 2-group structure, note that there are two edges with vertices along them: $v_2$ and $v_6$ respectively. We now excise these in turn. Excision of the vertex $v_2$ (and thereby the edge $e = 1$), results in $\mathcal{E}_{e=1} = \mathbb{Z}_4$, so this participates in a 2-group, whereas excision of $v_6$ results in $\mathcal{E}_{e=4} = \mathbb{Z}_2$, and this flavor symmetry will not participate in the 2-group. The flavor symmetry group of this theory is

$$\mathcal{F} = SO(3) \times SU(2) \tag{4.6}$$

and the first factor takes part in a non-trivial 2-group structure.

Using the brane-web approach we find the following brane-charges

$$W = \begin{pmatrix} 1 & 1 \\ 1 & 1 \\ -1 & 1 \\ -3 & -1 \\ 1 & -1 \\ 1 & -1 \end{pmatrix}, \tag{4.7}$$

from where we find the 1-form symmetry $\Gamma^{(1)} = \mathbb{Z}_2$. The flavor symmetry algebra can be read off from the toric diagram to be

$$\mathfrak{f} = \mathfrak{su}(2)^{(1)} \oplus \mathfrak{su}(2)^{(2)}, \tag{4.8}$$

where the first is associated to the multiplicity $m_{(-1,1)} = 2$ and the second to $m_{(1,1)} = 2$. From the reduced W-matrices we obtain

$$\mathrm{SNF}\left(W^{\mathrm{red}}_{(-1,1)}\right) = \mathrm{diag}(1,4), \qquad \mathrm{SNF}\left(W^{\mathrm{red}}_{(1,1)}\right) = \mathrm{diag}(1,2), \tag{4.9}$$

so that there is a non-trivial extension, where however only the first $\mathfrak{su}(2)$ factor participates in.

## 4.3 Non-Lagrangian Theories with 2-Groups

We now construct non-Lagrangian theories with 2-group symmetries. Consider the toric diagrams defined by [25, 67]:

$$B_N^{(2)}: \qquad ((N,0,1),(0,N-1-k,1)), \ k = 0, \cdots, N-2, \tag{4.10}$$

which have one-form symmetry $\Gamma^{(1)} = \mathbb{Z}_N$ and flavor symmetry group $\mathcal{F} = SU(N-2)/\mathbb{Z}_{N-2}$. In figure 7 we show the example for $B_4^{(2)}$. In this case we find that indeed

$$B_4^{(2)}: \qquad \Gamma^{(1)} = \mathbb{Z}_{\gcd(4,4,8)} = \mathbb{Z}_4, \tag{4.11}$$

and the 2-group symmetry arises after excising the $SO(3)$ flavor node. In general we find:

$$B_N^{(2)}: \qquad \Gamma^{(1)} = \mathbb{Z}_{\gcd(N,N,(N-2)N)} = \mathbb{Z}_N. \tag{4.12}$$

Excising the flavor vertices associated to $\mathcal{F}$ we find

$$\mathcal{E} = \mathbb{Z}_{N(N-2)}, \tag{4.13}$$

and thus there is a non-trivial extension only for $N = 2n$.

## 4.4 Generalized Toric Geometry

Although the precise geometric meaning of generalized toric diagrams (GTP) [60,68,69] still remains to be understood, we can nevertheless apply our approach to the dual brane-webs. Consider a GTP,

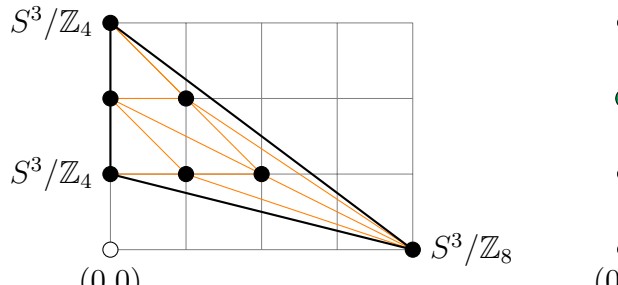 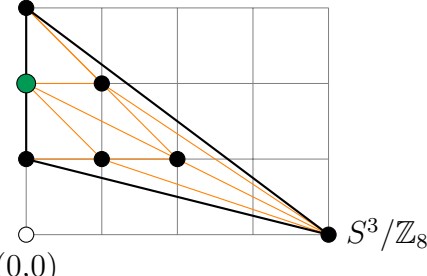

Figure 7: The toric diagram from $B_4^{(2)}$.

and let $P$ be the toric polygon where all white-dots are replaced by black dots. Let $W$ again be the set of $(p,q)$ brane-charges associated to $P$ (i.e. not the GTP), then the 1-form symmetry is computed from the SNF of $P$ [30]. Let $W$ be the set brane-charges corresponding to $P$.

Conjecturally we find the following rule: To compute the 1-form symmetry, compute as before in the toric case $\mathrm{SNF}(W)$. For the 2-group, we again excise the $(p,q)$-charges which have non-trivial multiplicities in the original GTP (not the one where all white dots have been replaced with black). Consider first the theory $SU(4)_0 + 1\boldsymbol{AS}$, which is closely related to the toric quivers we discussed earlier. It has $\Gamma^{(1)} = \mathbb{Z}_2$ and GTP

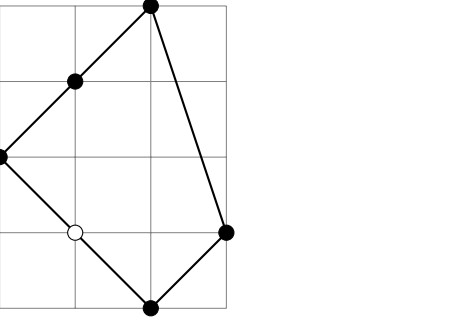

$$(4.14)$$

The associated brane-charges $W^{\mathrm{GTP}}$ for the GTP, and $W$-matrix for the "filled" web are

$$W^{\mathrm{GTP}} = \begin{pmatrix} 2 & 2 \\ -1 & 1 \\ -3 & -1 \\ 1 & -1 \\ 1 & -1 \end{pmatrix}, \qquad W = \begin{pmatrix} 1 & 1 \\ 1 & 1 \\ -1 & 1 \\ -3 & -1 \\ 1 & -1 \\ 1 & -1 \end{pmatrix} \qquad (4.15)$$

The only multiplicity that does not correspond to a white-dot is $m_{(1,1)} = 2$ and

$$\mathrm{SNF}(W_{(1,1)}^{\mathrm{red}}) = \mathrm{diag}(1,2)\,, \qquad (4.16)$$

so there is no 2-group, consistent with [20].

On the other hand $SU(4)_2 + 1\boldsymbol{AS}$ has $\Gamma^{(1)} = \mathbb{Z}_2$ and GTP

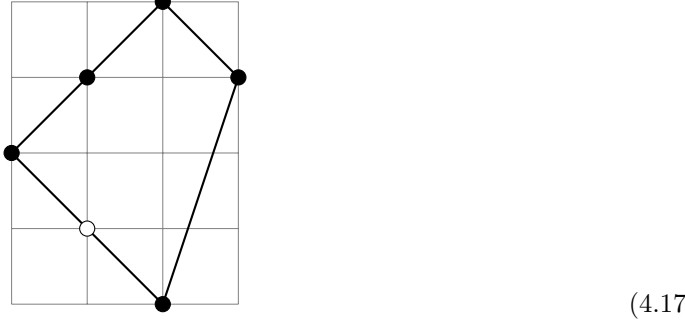

$$(4.17)$$

with

$$W^{\mathrm{GTP}} = \begin{pmatrix} 2 & 2 \\ -3 & 1 \\ -1 & -1 \\ 1 & -1 \\ 1 & -1 \end{pmatrix}, \qquad W = \begin{pmatrix} 1 & 1 \\ 1 & 1 \\ -3 & 1 \\ -1 & -1 \\ 1 & -1 \\ 1 & -1 \end{pmatrix} \tag{4.18}$$

Again the only non-white dot multiplicity is $m_{(1,1)} = 2$ but now

$$\mathrm{SNF}(W^{\mathrm{red}}_{(1,1)}) = \mathrm{diag}(1, 4), \tag{4.19}$$

indicating a non-trivial extension, again consistent with [20]. Clearly applying this to further toric and generalized toric models is straight forward, and only requires the flavor symmetry to be manifest in the geometric description.

## Acknowledgements

We thank Fabio Apruzzi, Lakshya Bhardwaj, Antoine Bourget, Jonathan Heckman, Saghar S. Hosseini, Max Hubner, Dave Morrison, Mark Powell and Yi-Nan Wang for discussions. SSN is supported in part by the European Union's Horizon 2020 Framework: ERC grant 682608 and in part by the "Simons Collaboration on Special Holonomy in Geometry, Analysis and Physics" grant number 724073. MDZ acknowledges support from the European Research Council (ERC) under the European Union's Horizon 2020 research and innovation programme (grant agreement No. 851931). MDZ and IGE are supported in part by the "Simons Collaboration on Global Categorical Symmetries." IGE is further supported in part by STFC grant ST/T000708/1.

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
