# Peer review of "-Group Symmetries and M-Theory"

_SciPost Physics_

## Round 1 · Referee Report · Anonymous (Referee 1) · 2022-6-27

Report

It is a nice paper where the authors explain how to extract the two-group structure formed by 1-form symmetries and 0-form flavor symmetries of 5d QFTs arising from M-theory on a 6-dimensional cone, from the geometry of the 5-dimensional link of the tip of the cone.

The exposition is clear with many examples, and the referee thinks that it can be published mostly as is. There are still some minor points the referee would like to raise, though:

  1. In Sec.2.1, the authors assume that $\mathcal{F}^{(0)}=F/C$ where $F$ is simply connected. Is it necessarily the case? In principle it can happen that what appears in the 2-group structure is not the simply-connected cover $F_\text{simply-connected}$ but a group $F$ intermediate between $\mathcal{F}^{(0)}$ and $F_\text{simply-connected}$, because not all of the representations of $F_\text{simply-connected}$ appear as the line-changing operators, and only the representations of $F$ do.

The authors can keep their definition of $F$ and $C$ as is in the manuscript; but then Eq.(2.3) needs to be modified to

$$ 0\to \ker{\alpha} \to \hat{\mathcal{E}}\stackrel{\alpha}{\longrightarrow}\hat{\Gamma}^{(1)}\to 0 $$
where $\ker\alpha$ is a subgroup of $\hat C$.

  1. Below Eq.(2.7), the authors seem to suggest that "both $w_2$ and $H^3$ nontrivial would mean $\mathrm{Bock}(w_2)\in H^3$ is nontrivial". But of course it can happen that a nontrivial $w_2$, a nontrivial $\mathrm{Bock}$ and a nonzero $H^3$ can still lead to $\mathrm{Bock}(w_2)=0 \in H^3$.

The referee understands that the authors understand this point very well and that they just used generally sloppy language of physical mathematics, but as the papers in this subject become increasingly mathematical the referee thinks that the authors can also try to be mathematically a bit more precise.

  1. The accent marks of Cvetič, Córdova and Schäfer are not consistently applied in the bibliography.

  2. Ref.[45] which "just appeared" should be given a preprint number at least.

---

## Round 1 · Referee Report · Anonymous (Referee 2) · 2022-7-17

Report

This paper discussed 2-group symmetries in the context of 5d theories constructed via geometric engineering from M theory. It provided a concrete algorithm to compute the 2-group symmetries from the geometric data and demonstrated the validate of the methods in various examples.

Overall, the referee thinks that the paper is very nice, and therefore recommend the publication of this manuscript.

One minor point:
1. In the paragraph below (2.12), there is a sentence starting with “An explicit analysis of the inclusion map $H_1(A\cap B) = H_1(A) \oplus H_1(B)$”. Should it be $H_1(A\cap B) \rightarrow H_1(A) \oplus H_1(B)$ instead?

---

## Round 2 · Author Response

We are very thankful to the referees for the very useful comments. We have addressed them in the revised version as follows:
Point 1 of Referee 1: We have modified the discussion between (2.2) and (2.4) to indicate that indeed ker\alpha is the general form but in our cases it is always \widehat{C}. We also agree that more generally F does not have to be the simply-connected version
but this is not a very strong assumption and a generalization to non-simply connected F is straightforward. The choice made in the paper makes the presentation less cluttered.
Point 2 of Referee 1: This was indeed very poorly phrased, and potentially misleading. We have added a paragraph on page 6 (just above
the beginning of the "2-Groups." paragraph), and some additional comments below (2.7) that should now be clearer.
We have also implemented the other comments by Referee 1 and Referee 2.
Point 1 of Referee 1: We have modified the discussion between (2.2) and (2.4) to indicate that indeed ker\alpha is the general form but in our cases it is always \widehat{C}. We also agree that more generally F does not have to be the simply-connected version
but this is not a very strong assumption and a generalization to non-simply connected F is straightforward. The choice made in the paper makes the presentation less cluttered.
Point 2 of Referee 1: This was indeed very poorly phrased, and potentially misleading. We have added a paragraph on page 6 (just above
the beginning of the "2-Groups." paragraph), and some additional comments below (2.7) that should now be clearer.
We have also implemented the other comments by Referee 1 and Referee 2.

---

## Editorial Decision

resubmitted